# LOCAL: Latent Orthonormal Contrastive Learning for Paired Images

## Abstract

Classification with comparative paired inputs, such as pre- and post-disaster satellite images, distinguishes classes of samples by encompassing dual feature sets that individually characterize a sample. Representation learning from comparative nature of the inputs calls for not only recognizing invariant patterns shared across all inputs but also effectively differentiating the contrastive attributes present between each pair of inputs. Supervised Contrastive Learning (SCL) aims to learn representation that maximally separates different classes and condenses within individual classes, thereby attaining an adversarial equilibrium. However, this equilibrium typically relies on the assumption of balanced data and large batch sizes for sufficient negative sampling. These issues are exacerbated when applied to paired satellite images due to increased computational load, high-resolution data, and severe class imbalance. To address these challenges, we introduce Latent Orthonormal Contrastive Learning (LOCAL), an approach that optimizes class representations in an orthonormal fashion. By learning each class to a unique, orthogonal plane in the embedding space, LOCAL is efficient with smaller batch sizes, provably effective regardless of class size imbalance, and yields more discriminative information between pairs of inputs via a feature correlation module. Experimental results on paired image data demonstrate superior performance of LOCAL over SCL, offering a powerful alternative approach for paired input analysis.

## 1 Introduction

Comparative paired input datasets consists of two paired inputs for each sample that are compared against each other. These pairs are distinguished by dual feature sets that individually characterize each sample. This form of data representation is particularly significant in fields that require the comparison of two related but distinct sets of data. One notable example is pre- and post-disaster damage assessment, where paired satellite images captured before and after a natural disaster (such as a hurricane, flooding, or earthquake) are compared, as shown in Fig. 1. Such comparative analysis enhances the effectiveness of damage classification, as the side-by-side analysis helps detect subtle changes and assess the extent of the damage more accurately Kamari & Ham (2022); Ma (2021); Cheng et al. (2021); Berezina & Liu (2022). This principle is similarly applied in fields like natural language inference (NLI), where models aim to understand the relationship between sentence pairs (e.g., premise and hypothesis)MacCartney & Manning (2008); Shen et al. (2022), and in medical imaging, where paired scans (such as pre- and post-treatment MRIs) are compared to track changes over time Kooi & Karssemeijer (2017); Bai et al. (2024); Perek et al. (2019), *etc.*

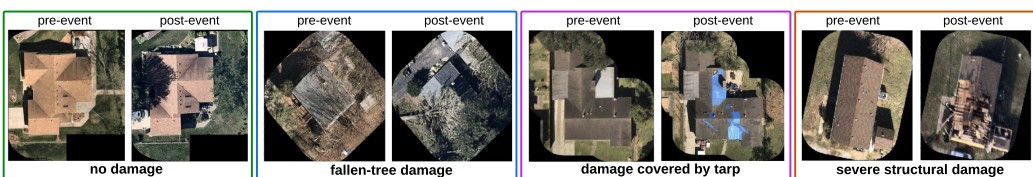

Figure 1: Comparative damage classification using pre- and post-disaster satellite imagery.

In recent years, self-supervised contrastive learning has emerged as a powerful technique across various domains, especially in computer vision Chen et al. (2020); He et al. (2020); Caron et al. (2020); Grill et al. (2020), yielding superior performance in representation learning. The general idea of contrastive learning is to train network models to pull together an anchor sample and a "positive" sample in the embedding space, while simultaneously pushing the anchor away from multiple "negative" samples. SimCLR Chen et al. (2020) is such an example that learns visual representations by constructing positive and negative samples without actual labels, leveraging data augmentations Cubuk et al. (2019; 2020). Additionally, Khosla et al. Khosla et al. (2020) extended it to the fully-supervised setting, proposing a supervised contrastive loss (SCL) that uses label information to align samples from the same class while separating those from different classes. Further theoretical analysis by Graf et al. Graf et al. (2021) examined SCL and the cross-entropy loss, showing that while both losses aim for a similar geometric solution in the embedding space, SCL converges much closer to the optimal target, leading to a better generalization performance. Graf et al.'s analysis showed that the optimal embeddings, when minimizing the loss, result in a single embedding for all points within a class, with per-class embeddings forming a *regular simplex* inscribed in the hypersphere, representing a highly efficient and well-separated geometric arrangement of the embeddings.

However, despite its advantages, SCL faces two critical and interconnected challenges: imbalanced classes and requirement of large batch sizes.

- First, the theoretical guarantee of SCL, as discussed by Graf et al. Graf et al. (2021), is contingent upon a critical assumption of balanced data, which is often unrealistic in real-world applications. When SCL is applied to imbalanced datasets, the resulting poor uniformity can significantly degrade model performance Cui et al. (2021); Kang et al. (2020); Li et al. (2022); Zhu et al. (2022); Wang et al. (2021). Classes of higher frequency have a greater lower bound on misclassification loss Cui et al. (2021), which skews the model toward these dominant classes and leads to biased representations. Then, SCL fails to form a regular simplex Zhu et al. (2022) in the embedding space. Instead, it forms an asymmetrical structure where high-frequency classes are more widely scattered, while low-frequency classes are drawn closer together, making it difficult for the model to learn robust and discriminative features for minority classes.
- Second, the effectiveness of contrastive learning relies on a rich set of negative samples to adequately separate representations in the embedding space. Ensuring sufficient negative diversity typically requires large batch sizes. Both theoretical insights and empirical evidence have shown that increasing the number of negative samples can significantly improve contrastive learning performance Bachman et al. (2019); Tian et al. (2020); Chuang et al. (2020); Wang & Isola (2020). However, this improvement comes at the cost of increased memory consumption, which poses significant challenges for resource-constrained computing environments. As a result, SCL often suffers from performance degradation when batch sizes are small.

The above challenges inherent to SCL are further exacerbated when applied to comparative paired input datasets, particularly in the context of damage assessment using satellite images, due to several key factors. First, paired inputs require the model to process both images for each sample simultaneously. This effectively doubles the computational workload, thus thus necessitating the use of smaller batch sizes due to memory constraints. Second, the high-resolution nature intensifies the issue. These images often contain thousands of pixels, significantly larger than the low-resolution images typically found in benchmarks like CIFAR-10 or CIFAR-100, further limiting batch sizes and reducing the diversity of negative samples in SCL. While resizing images can ease computational constraints issue, this sacrifices important details crucial for accurate analysis, ultimately degrading model performance. Third, the imbalance inherent in these datasets, where categories like "no damage" dominate, but minority classes like "severe structual damage" extremely underrepresented, adds considerable complexity. The smaller batch sizes required for processing paired, high-resolution images reduce the model's exposure to minority class examples, making it even harder to learn efficient representations for these underrepresented categories.

To overcome the aforementioned challenges as a whole, we introduce LOCAL (Latent Orthonormal Contrastive Learning) for paired images, optimizing the representations of distinct classes in an orthonormal fashion. In LOCAL, each class occupies a unique plane, and these planes are orthogonal to one another, enhancing class separation. We theoretically justify that this approach bypasses the need to construct a regular simplex, as required in SCL, and alleviates the assumption of balanced

data without relying on large batch sizes. Additionally, we incorporate a feature correlation module to capture hierarchical features from intermediate layers, further improving joint representation learning between paired inputs. The key contributions of this work are as follows:

- We propose a novel approach to construct novel orthonormal embeddings for different classes, rather than mapping distinct classes to vertices of a regular simplex inscribed in a hypersphere as in SCL. This enhances the discriminative power between classes and requires only minor adjustments to standard SCL code.

- By eliminating the dependency on adversarial equilibrium, our method allows for the use of smaller mini-batches, which is crucial when working with paired input settings and high-resolution images, effectively addressing computational constraints.

- Our theoretical analysis proves a lower bound for the proposed new contrastive loss function and shows that minimizing this new loss reaches the lower-bound regardless of the level of class balance.

- We incorporate a feature correlation module that utilizes latent hierarchical features derived from intermediate layers to enhance joint representation learning between paired inputs. This is integrated into a broader framework that combines representation learning with classification.

## 2 PRELIMINARIES AND MOTIVATIONS

Our approach has been particularly motivated by the situation of large sample of paired inputs but with extremely skewed class distribution where we observe notable challenges for the application of SCL, particularly in the context of damage assessment using high-resolution satellite imagery,

**Regualr Simplex in SCL.** The supervised contrastive loss can be minimized to train an encoder, which is designed to attract pairs of samples from the same class (referred to as *positives*) and push away pairs of samples from different classes (referred to as *negatives*), spatially separating balanced classes to the maximal ex-

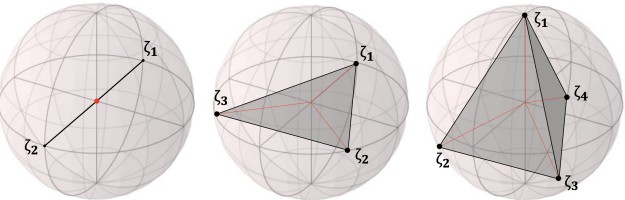

(a) 1-simplex, $K{=}2$ (b) 2-simplex, $K{=}3$ (c) 3-simplex, $K{=}4$

Figure 2: Origin-centered regular simplices inscribed in $S^2_{\rho=1}$

tent, thereby attaining an adversarial equilibrium. Graf et al. Graf et al. (2021) have shown that the distribution of optimal embeddings obtained by minimizing the loss has only a single embedding for points in a class, with the per-class embeddings collectively forming a *regular simplex* inscribed in the hypersphere, denoted as $S^{h-1}_\rho$ in $h$-dimensional space with radius $\rho$. Let $K$ denote the number of classes, it is proved that supervised contrastive learning (SCL) attains its minimum if and only if the representations of each class collapse to the vertices of an origin-centered regular $K-1$ simplex. Let $\zeta_1, \ldots, \zeta_K \in \mathbb{R}^h$ be the vertices of a regular simplex, satisfying 1) $\sum_{i \in [K]} \zeta_i = 0$; 2) $\|\zeta_i\| = 1$ for $i \in [K]$; 3) $\exists d \in \mathbb{R} : d = \langle \zeta_i, \zeta_j \rangle$ for $i \neq j$. Fig. 2 demonstrates the vertices for $K = 2, 3, 4$ on the unit hypersphere $S^2_{\rho=1}$ in three dimensional space.

**Batch Size Limitations and Memory Bottlenecks in Paired Input Learning.** One of the key challenges in SCL is is the preference for large batch sizes to stabilize optimization during training. This becomes a significant issue when working with paired inputs, particularly for high-resolution images which typically involve larger data samples. The need to process two images per sample effectively doubles the data load, while the higher resolution further limits the feasible batch sizes that can be used due to memory constraints. On resource-constrained hardware, such as edge devices or GPUs with limited memory capac-

| Dataset | BS | InputSize | GPU Memory Usage |
|---------|-----|-----------|------------------|
| HRA | 8 | 512×512 | 9753 (9.5 G) |
| | 16 | 512×512 | 17495 (17 G) |
| | 8 | 256×256 | 4303 (4.2 G) |
| | 16 | 256×256 | 6153 (6 G) |

Table 1: GPU memory usage (MB) comparison for batch sizes (BS) and downsized input resolutions (paired)

ity, managing these batch sizes with larger data samples becomes increasingly difficult, creating a bottleneck for efficient training. For instance, as shown in Table 1, the memory footprint for process-

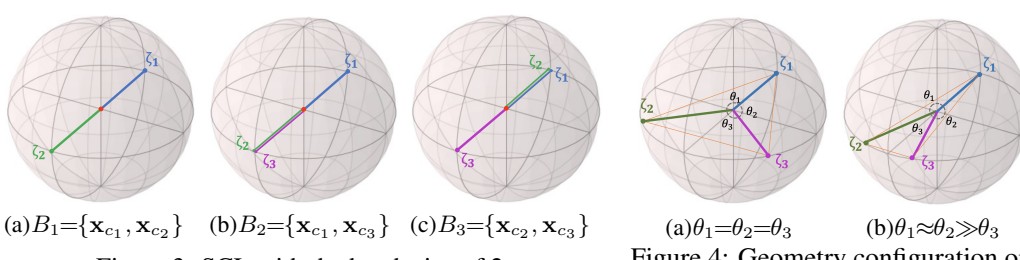

(a)$B_1{=}\{\mathbf{x}_{c_1}, \mathbf{x}_{c_2}\}$   (b)$B_2{=}\{\mathbf{x}_{c_1}, \mathbf{x}_{c_3}\}$   (c)$B_3{=}\{\mathbf{x}_{c_2}, \mathbf{x}_{c_3}\}$     (a)$\theta_1{=}\theta_2{=}\theta_3$    (b)$\theta_1{\approx}\theta_2{\gg}\theta_3$

Figure 3: SCL with the batch size of 2.     Figure 4: Geometry configuration on balanced or imbalanced data

ing even a very small batch size of 16 can range from 4.2 GB to 17 GB of GPU memory, depending on the input resolution (already downsized from the original), making it difficult to achieve efficient training without running into memory constraints. This leads to a trade-off between downsizing the inputs to reduce memory consumption and the risk of losing critical information necessary for accurate analysis.

Smaller batch sizes introduce instability into the optimization process for SCL, by providing fewer negative samples which may cause embedding shifting. To illustrate this, consider the following simplified example. Without loss of generality, assume a mini-batch size of $|B| = 2$, meaning that *at most* two classes are optimized in each batch. Suppose we have the following batches: $B_1 = \{\mathbf{x}_{c_1}, \mathbf{x}_{c_2}\}$, $B_2 = \{\mathbf{x}_{c_1}, \mathbf{x}_{c_3}\}$, and $B_3 = \{\mathbf{x}_{c_2}, \mathbf{x}_{c_3}\}$, where $\mathbf{x}$ denotes a sample (can be single or paired settings), and $c_i$ indicates that the sample $\mathbf{x}_{c_i}$ belongs to the $i$-th class. As shown in Fig. 3, during optimization for $B_1$, $\zeta_1$ and $\zeta_2$ form a *1-simplex* at optimality. However, when $B_2$ comes in, $\zeta_3$ is pushed away from $\zeta_1$, creating a situation where $\zeta_3$ and $\zeta_2$ risk collapsing. When $B_3$ arrives, $\zeta_3$ and $\zeta_2$ repel, but $\zeta_1$ and $\zeta_2$ (or $\zeta_3$) collapse. This cyclical collapse causes embedding drift, preventing convergence and breaking equilibrium. The root cause is that there are unique simplex vertices to which per-class images can map. In contrast, we argue that representations of different classes can map into orthogonal subspaces in our proposed method, which reduces the risk of embedding drift in SCL (as illustrated Fig. 5 in later Section 3.3).

**Imbalanced Data.** It has been proved that SCL reaches its ideal geometry configuration for representation learning when it achieves its minimum on a *balanced* data batch Graf et al. (2021). This is illustrated in Fig. 4(a), where the angles between the embeddings of distinct classes are equal and maximized, e.g., $\theta_1 = \theta_2 = \theta_3 = 2\pi/3$ for $K = 3$. However, optimizing the SCL may fail to form a regular simplex for imbalanced long-tailed data Zhu et al. (2022). In such scenarios, high frequency classes dominate the learning process Cui et al. (2021); Kang et al. (2020), leading to unequal angles between embeddings, as depicted in Fig. 4(b), where $\theta_1 \approx \theta_2 \gg \theta_3$. This imbalance can bias the model towards majority classes, potentially resulting in suboptimal performance for the minority classes. Existing strategies include balancing the number of positive samples across all classes within each batch Kang et al. (2020), introducing a set of class-wise learnable centers to rebalance from an optimization perspective Cui et al. (2021), incorporating a classifier branch to eliminate the bias of the classifier towards head classes Wang et al. (2021), assigning pre-computed uniformly distributed targets to each class prior to training Li et al. (2022), or optimizing all classes to a balanced feature spaceZhu et al. (2022).

In real-world scenarios such as paired inputs for high-resolution satellite imagery, these imbalances can be even more pronounced. The need for limited batch sizes makes it even harder to properly handle minority classes, increasing the importance of developing more robust contrastive learning frameworks. While the existing methods aim to maintain the regular simplex in the SCL framework under imbalanced data conditions, this simplex requirement may constrain the model's flexibility. We argue that alternative geometric approaches may offer more effective solutions for handling imbalanced data in such complex scenarios.

# 3 LOCAL: Latent Orthonormal Contrastive Learning Framework

## 3.1 Decoupling Positives and Negatives from SCL

The SCL loss is derived first in Khosla et al. (2020) by extending the self-supervised contrastive loss to take label information. In a mini-batch $B$ consistingof training samples $\{\mathbf{x}_i, y_i\}_{i=1}^{|B|}$, each sample

$\mathbf{x}_i$ is represented as a pair of inputs $(\mathbf{x}_i^{pre}, \mathbf{x}_i^{post})$, corresponding to pre- and post-event images, with $y_i$ being the associated class label. *Positives* are defined as those samples with the same class label as $\mathbf{x}_i$, while *negatives* belong to different classes. Let $\{\mathbf{z}_i\}^{|B|}i = 1$ denote the set of embedding features, where $\mathbf{z}_i$ contains the embeddings of both elements in the paired input $\mathbf{x}_i$. SCL is formulated as:

$$\mathcal{L}_{SCL} = \frac{1}{|B|} \sum_{i \in B} \frac{-1}{|B_{y_i}| - 1} \sum_{p \in B_{y_i} \setminus \{i\}} \mathcal{L}^{SCL}(\mathbf{z}_i), \;\; \text{where } \mathcal{L}^{SCL}(\mathbf{z}_i) = \log \frac{\exp(\langle \mathbf{z}_i, \mathbf{z}_p \rangle / \tau)}{\sum_{a \in B \setminus \{i\}} \exp(\langle \mathbf{z}_i, \mathbf{z}_a \rangle / \tau)}$$

Here, $\langle \cdot, \cdot \rangle$ denotes the inner product, $\tau \in \mathbb{R}^+$ is the scalar temperature parameter and we omit $\tau$ in the subsequent sections for simplicity. Without loss of generality, $B_{y_i} \equiv \{p \in B : y_p = y_i\}$ denote the set of indices in the batch $B$ with label equal to $y_i$, and $|B_{y_i}|$ is its cardinality. We decouple the *positives* and *negatives* in the denominator and the term $\mathcal{L}^{SCL}(\mathbf{z}_i)$ can be re-written as:

$$\mathcal{L}^{SCL}(\mathbf{z}_i) = \log \frac{\exp(\langle \mathbf{z}_i, \mathbf{z}_p \rangle)}{\sum_{p \in B_{y_i} \setminus \{i\}} \exp(\langle \mathbf{z}_i, \mathbf{z}_p \rangle) + \sum_{n \in B_{y_i}^C} \exp(\langle \mathbf{z}_i, \mathbf{z}_n \rangle)} \tag{1}$$

where $B_{y_i}^C$ is the complementary set of $B_{y_i}$ such that $B_{y_i} + B_{y_i}^C = B$, including indices of all negatives in the batch $B$. Eq.(1) encourages the feature representations from positive pairs to be similar but negative pairs to be dissimilar. The loss attains its minimum once the representations of each class collapse to the vertices of a regular simplex, inscribed in a unit hypersphere Graf et al. (2021)..

## 3.2 Orthonormal Contrastive Loss

To alleviate the issues described in Section 2, we propose the new orthonormal contrastive loss (OCL) with the following loss function:

$$\mathcal{L}^{OCL}(\mathbf{z}_i) = \log \frac{\exp(\langle \mathbf{z}_i, \mathbf{z}_p \rangle)}{\sum_{p \in B_{y_i} \setminus \{i\}} \exp(\langle \mathbf{z}_i, \mathbf{z}_p \rangle) + \sum_{n \in B_{y_i}^C} \exp(|\langle \mathbf{z}_i, \mathbf{z}_n \rangle|)} \tag{2}$$

The key difference between Eq. (equation **??**) and Eq. (equation 1) is that, in OCL, negatives are not simply pushed way from the anchor $z_i$, but instead are made perpendicular to the anchor's embedding. While SCL maps per-class embeddings to the vertices of a regular simplex, OCL aims to learn pairwise perpendicular subspaces for each class. OCL follows a similar logic as SCL—attraction and repulsion—but decouples the intra-class and inter-class forces differently. more explicitly. The similarity between $\mathbf{z}_i$ and $\mathbf{z}_j$ is commonly measured using the *cosine* similarity: $S_{i,j} = \frac{\langle \mathbf{z}_i, \mathbf{z}_j \rangle}{\|\mathbf{z}_i\| \|\mathbf{z}_j\|} = \langle \mathbf{z}_i, \mathbf{z}_j \rangle$ if the embeddings are normalized to unit vectors. In this context, OCL optimizes the attraction within a class by maximizing cosine similarity, $S_{i,p} = \langle \mathbf{z}_i, \mathbf{z}_p \rangle \in [-1, 1]$, ensuring that embeddings of positive samples are pulled closer together. Simultaneously, it enforces orthogonality between classes by minimizing $S_{i,n} = |\langle \mathbf{z}_i, \mathbf{z}_n \rangle| \in [0, 1]$ for negative samples, driving them toward perpendicularity. Through this decoupling of intra-class attraction and inter-class repulsion, OCL provides an alternative geometric solution for supervised contrastive learning.

## 3.3 Theoretical Analysis

We establish a lower-bound for the proposed OCL to construct orthonormal learned embeddings without contingency on data balance constraints. Assuming that the encoder has sufficient expressive capability, a lower bound on the SCL loss is derived as follows in Graf et al. (2021). Let $N$ be the total number of examples, $K$ the total number of different classes, and $D_E$ the embedding dimension.

**Theorem 3.1.** *Let $\mathcal{Z} = S^{D_E - 1} = \{\mathbf{z} \in \mathbb{R}^{D_E} : \|\mathbf{z}\| = 1\}$, and $Z = \{\mathbf{z}_i \mid \forall i \in [N], \mathbf{z}_i \in \mathcal{Z}\}$ be an $N$ point configuration with labels $Y = \{y_i \mid \forall i \in [N], y \in [K]\}$. If the label configuration $Y$ is balanced, for any class $y$ and any batch $B$, the class-specific batch-wise loss is bounded by*

$$\mathcal{L}_{SCL}(Z; Y) \geq \sum_{l=2}^{|B|} l M_l \log \left( l - 1 + (|B| - l) \exp(\frac{-K}{K-1}) \right). \tag{3}$$

*where $M_l = \sum_{y \in [K]} \{B \in \mathcal{B} \mid |B_y| = l\}$, $\mathcal{B} = \{\{n_1, n_2, \ldots, n_{|B|}\} \mid n_i \in [N], \forall i \in [B]\}$ is the set of all index multi-sets of size $|B|$, and the set $B_y$ consists of all samples with label $y$ in the batch $B$. Equality is attained if and only if the following conditions are satisfied. There are*

$\zeta_1, \zeta_2, \ldots, \zeta_K \in \mathbb{R}^{D_E}$ *with a large $D_E$, s.t. $K < D_E + 1$ such that:*

*C1) $\forall n \in [N]$: $\mathbf{z}_n = \zeta_{y_n}$;*

*C2) $\{\zeta_y\}_y$ form a regular simplex.*

Theorem 3.1 implicitly suggests $|B|$ to be large so as to achieve the adversarial equilibrium, where each class stays away from other classes to the maximal extent. It is also critical to have balanced data for SCL. SCL can fail with long-tailed data due to intra-class feature collapse and inter-class uniformity issues dominated by classes with higher frequencies Zhu et al. (2022). Our OCL loss function can mitigate the impact of data imbalance on the repulsion term, by allowing more options than simplex vertices. A lower bound of the OCL loss is characterized by Theorem 3.2.

**Theorem 3.2.** *Let $Z$ and $Y$ be defined as in **Theorem 3.1**, without the assumption that the label configuration Y is balanced. We have*

$$\mathcal{L}_{OCL}(Z;Y) \geq \sum_{l=2}^{|B|} l M_l \log \left( l - 1 + \frac{|B| - l}{e} \right). \tag{4}$$

*Equality is attained if and only if there are $K$ orthonormal vectors $\xi_1, \xi_2, \ldots, \xi_K \in \mathbb{R}^{D_E}$ with a value of $D_E$, s.t. $K < D_E$ can be obtained under the condition that:*

*C1) $\forall n \in [N] : \mathbf{z}_n = \xi_{y_n}$.*

*Proof. See the Supplementary Material A.1.*

**Discussion.** As shown in Fig. 5, according to **Theorem 3.2**, optimizing the OCL loss would not make models trap in the settings described in Section 2. For $B_1 = \{\mathbf{x}_{c_1}, \mathbf{x}_{c_2}\}$, rather than having only one single option for $\zeta_2$ in SCL, which seeks to form a *1-simplex* with $\zeta_1$, the OCL loss automatically expands the search space for its optimal solution $\xi_2$, from a deter-

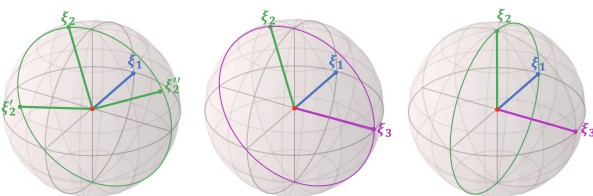

(a)$B_1 = \{\mathbf{x}_{c_1}, \mathbf{x}_{c_2}\}$    (b)$B_2 = \{\mathbf{x}_{c_1}, \mathbf{x}_{c_3}\}$    (c)$B_3 = \{\mathbf{x}_{c_2}, \mathbf{x}_{c_3}\}$

Figure 5: OCL with the batch size of 2.

mined goal to unlimited options within the plane (in green color) perpendicular to $\xi_1$ (e.g., $\xi_2, \xi_2', \xi_2''$). When it comes to $B_2 = \{\mathbf{x}_{c_1}, \mathbf{x}_{c_3}\}$, $\xi_3$ will also be projected to be orthogonal to $\xi_1$ and falls in the same plane, while it is highly unlikely that $\xi_3$ will collapse with $\xi_2$. When $B_3$ arrives, $\xi_3$ and $\xi_2$ are repelled to be orthogonal via the OCL loss, with minimal effect on $\xi_1$. Therefore, our method offers an advantage on memory consumption - during OCL training, it reaches the lower bound as long as the representation from different classes become orthogonal, so it does **not rely on large batch size** to include samples from all classes so as to achieve stable adversarial equilibrium between all classes as shown in standard SCL. It is also noteworthy that Theorem 3.2 suggests that our loss function can reach the lower-bound **without being constrained to the level of data balance** unlike Theorem 3.1. With this theorem, we justify that minimizing this new loss would not force embeddings of different minority classes to collapse to the same vertices. The OCL method assumes that the angle between a dominant class and other classes is orthogonal, corresponding to independent bases in the hyperspace.

### 3.4 END-TO-END LEARNING OF PAIRED INPUTS

Fig. 6 shows the overview framework for learning both representation and classification based on paired inputs. It includes two parts: the first part learns a feature mapping with the property of intra-class compactness and inter-class separability; whereas the second part is expected to learn a less biased classifier based on the orthonormal representations produced by the first part. We take the damage detection task as an example where pre- and post-disaster image pair is denote by $(\mathbf{x}^{pre}, \mathbf{x}^{post})$.

**Encoder Network**, $Enc(\cdot)$, can employ any suitable backbone network, e.g., ResNetHe et al. (2016), and maps either image $\mathbf{x}^{pre}$ and $\mathbf{x}^{post}$ in the pair to a vector representation, $\mathbf{r}^{pre} = Enc(\mathbf{x}^{pre}) \in \mathbb{R}^{D_R}$ and $\mathbf{r}^{post} = Enc(\mathbf{x}^{post}) \in \mathbb{R}^{D_R}$, whereas $\mathbf{r}^{pre}$ and $\mathbf{r}^{post}$ are normalized to be on the unit hypersphere in $\mathbb{R}^{D_R}$.

**Latent Hierarchical Feature Correlation Module**, $Lat(\cdot)$, is a module injected into the backbone network to learn latent hierarchical joint representation between $\mathbf{x}^{pre}$ and $\mathbf{x}^{post}$. Specifically, from

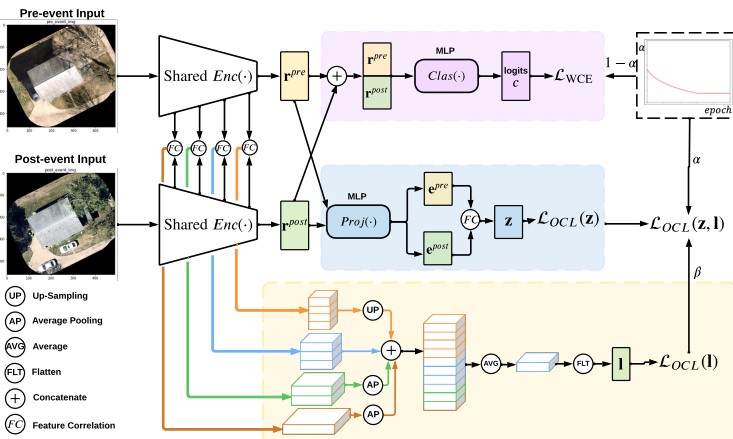

Figure 6: The end-to-end learning of the **LOCAL** model.

the backbone network, we firstly extract the multi-scale outputs of each block (e.g., four blocks for ResNet) and then the feature correlation between the pre- and post-event outputs from each block. Denote the output from each block as $Enc_i'(\cdot)$, $i \in [1, ..., 4]$. Our feature correlation module can be computed as $Enc_i'(\mathbf{x}^{pre}), Enc_i'(\mathbf{x}^{post}))) = \mathbf{W}_i([Enc_i'(\mathbf{x}^{pre}), Enc_i'(\mathbf{x}^{post})])$ where the matrix $\mathbf{W}_i \in \mathbb{R}^{d_i \times 2d_i}$ denotes the correlation parameters, and $[Enc_i'(\mathbf{x}^{pre}), Enc_i'(\mathbf{x}^{post})] \in \mathbb{R}^{2d_i}$ is the concatenated vector of $Enc_i'(\mathbf{x}^{pre}), Enc_i'(\mathbf{x}^{post}) \in \mathbb{R}^{d_i}$. Practically, $\mathbf{W}_i$ is set to $[\mathbf{I}_{d_i}, -\mathbf{I}_{d_i}]$ to compute the difference between pre- and post-event outputs, yielding satisfactory results. Then the feature correlation maps (lower left part of Fig. 6) illustrate the variation in dual images. These maps are resized to the same size via average pooling and up-sampling, and concatenated to form a hierarchical latent feature maps. Then, averaging over all channels produces a single channel feature map, which is then flattened and normalized to give a latent feature embedding $\mathbf{l} = Lat(\mathbf{x}^{pre}, \mathbf{x}^{post}) \in \mathbb{R}^{D_L}$. This embedding incorporates latent supervision from the backbone network, and then contributes to the computation of latent orthonormal contrastive loss $\mathcal{L}_{OCL}(\mathbf{l})$.

**Projection Network with Feature Correlation Module**, $Proj(\cdot)$ maps $\mathbf{r}^{pre}$ and $\mathbf{r}^{post}$ to the corresponding embedding vectors $\mathbf{e}^{pre} = Proj(\mathbf{r}^{pre}) \in \mathbb{R}^{D_E}$ and $\mathbf{e}^{post} = Proj(\mathbf{r}^{post}) \in \mathbb{R}^{D_E}$. This network is a multi-layer perceptron (MLP) with a hidden layer and an output layer of size $D_E$. It has been shown that such a projection module improves the quality of the embeddings of the layers preceding it Khosla et al. (2020); Chen et al. (2020). We apply an $\ell_2$ normalization to $\mathbf{e}^{pre}$ and $\mathbf{e}^{post}$ to ensure that the inner product can be used as the *cosine* similarity measure. A similar feature correlation module is incorporated to learn the variation between the pre- and post- outputs of the projection network. $\mathbf{z} = \mathbf{W}([\mathbf{e}^{pre}, \mathbf{e}^{post}])$ is used to compute the OCL loss $\mathcal{L}_{OCL}(\mathbf{z})$.

In our framework, the proposed OCL takes effect on both the latent hierarchical feature embedding $\mathbf{l}$ and the joint embbedding of the paired inputs $\mathbf{z}$, leading to the latent OCL loss:

$$\mathcal{L}_{OCL}(\mathbf{z}, \mathbf{l}) = \mathcal{L}_{OCL}(\mathbf{z}) + \beta\mathcal{L}_{OCL}(\mathbf{l}) \tag{5}$$

where $\beta \geq 0$ is a hyperparameter for tuning and $\mathcal{L}_{OCL}(\mathbf{l})$ can be regarded as a regularizer which regularizes the orthonormality class representations of lower-level features (with high contrast) that are extracted by the deep neural network. Imposing this regularizer helps learn the final inter-class orthonormal embeddings.

**Classification Network**, $Clas(\cdot)$, takes in the concatenated representation, $concat(\mathbf{r}^{pre}, \mathbf{r}^{post})$, from the *Encoder Network*. A non-linear MLP with a hidden layer and an output layer of the class size is employed to predict the class-wise logit values $\mathbf{c} \in \mathbb{R}^{D_C}$ of the input image pair, which are used to compute the weighted cross-entropy (WCE) loss $\mathcal{L}_{WCE}$. Combining with the WCE loss for classifier learning where the weight is the reciprocal of the appearance frequency of each class, we arrive at our final loss function of our proposed LOCAL: Latent Orthonormal Contrastive Learning Framework:

$$Total\ loss = \alpha\mathcal{L}_{OCL}(\mathbf{z}, \mathbf{l}) + (1 - \alpha)\mathcal{L}_{WCE} \tag{6}$$

where $0 \leq \alpha \leq 1$ is a weighting coefficient inversely proportional to the number of epochs.

## 4  EXPERIMENTS

### 4.1  HIGH-RESOLUTION PAIRED SATELLITE IMAGERY ANALYSIS

#### 4.1.1  EXPERIMENTAL SETUP

**Datasets.** We evaluate the performance and effectiveness of LOCAL on two remote sensing datasets with paired satellite images: the **HRA** dataset, a smaller-scale dataset containing 3,389 image pairs across 5 classes, and the large-scale public **xBD** dataset Gupta et al. (2019) encompassing 67,782 image pairs distributed across 4 classes. Both datasets include pre-disaster and post-disaster paired satellite images, allowing us to assess the model's capability in accurately predicting different types of damage across diverse geographical and disaster scenarios.

**Baselines.** To ensure fair comparison in our paired input setting, we extend original Supervised Contrastive Learning (SCL) into two variants: (1) the first variant combines SCL with WCE, and we refer to this as *SCL* for simplicity by omitting WCE; and (2) variant builds upon SCL by adding the latent hierarchical feature correlation module, refered to as *L-SCL*. Please refer to Appendix A.2 for detailed implementations to compare with our proposed method illustrated in Fig. 6. We compare these two variants with our proposed method, *LOCAL*. The main distinction between *SCL* and *L-SCL* lies in the inclusion of the latent hierarchical feature correlation module, represented as the yellow block in Fig. 6. The key difference between *L-SCL* and our method *LOCAL* is the use of the OCL loss instead of the SCL loss, which demonstrates the effectiveness of our proposed OCL loss.

**Working with Small Batch Sizes.** All algorithms are implemented in PyTorch and tested on servers equipped with NVIDIA A10 Tensor Core GPU with 24GB of GPU memory. According to Table 1, using a batch size of 16 and resizing the original high-resolution images to $512 \times 512$ occupied 17GB of GPU memory for the HRA dataset. To optimize memory usage, we carefully balanced the batch size and image downsizing to work within this limit.

#### 4.1.2  EVALUATION RESULTS

**Robustness to the Impact of Batch Size and Image Resizing.** Table 2 compares the three methods using a ResNet-50 as the backbone encoder. We conducted multiple 5-fold cross-validations for each method. Given the Limited GPU memory, we adjusted the batch sizes and image downsizing scales to ensure similar GPU memory usage across different experimental con-

| Dataset | BS | InputSize | *SCL* | *L-SCL* | *LOCAL* |
|---------|-----|-----------|-------|---------|---------|
| HRA | 8 | 512×512 | 76.47 (74.01-79.00) | 79.07 (77.15-81.55) | **82.78** (81.90-84.57) |
| | 16 | 512×512 | 78.49 (76.80-80.16) | 77.98 (76.57-80.28) | **83.64** (81.90-84.69) |
| | 8 | 256×256 | 75.87 (74.36-77.03) | 78.40 (77.03-79.58) | **81.76** (80.63-83.29) |
| | 16 | 256×256 | 77.26 (77.03-77.61) | 79.58 (78.18-81.90) | **81.20** (79.46-82.83) |
| xBD | 8 | 224×224 | 80.08 (79.57-80.50) | 80.98 (79.56-81.45) | **81.52** (80.26-83.41) |
| | 16 | 224×224 | 82.20 (81.48-83.88) | 81.09 (79.19-82.63) | **82.83** (81.83-83.18) |
| | 64 | 224×224 | 81.64 (81.17-82.07) | 79.05 (78.04-79.56) | **82.87** (81.88-84.32) |
| | 8 | 128×128 | 80.83 (80.24-81.73) | 81.01 (79.91-81.76) | **82.00** (81.42-83.08) |
| | 16 | 128×128 | 81.44 (81.02-81.78) | 80.56 (79.13-82.18) | **82.29** (80.46-83.28) |
| | 64 | 128×128 | 82.08 (80.93-82.68) | 79.45 (78.45-80.16) | **82.78** (82.11-83.59) |
| | 128 | 128×128 | 81.23 (80.86-81.75) | 79.57 (78.52-80.98) | **82.82** (82.06-83.94) |

Table 2: Performance comparison under variant batch sizes and image resizing scales

figurations. We observe that the available batch sizes are relatively small, typically around 8 or 16 for HAR. The general trend observed across varying batch sizes and image resizing scales shown in the column of *LOCAL* supports our hypothesis: larger batch sizes outperform smaller ones at the same downsizing scale, and larger image sizes perform better as they preserve more critical information.

The results demonstrate that *LOCAL* consistently outperforms the extended variants of *SCL* for paired input on both datasets. By comparing *L-SCL* with *SCL*, we validate the effectiveness of our proposed Latent Hierarchical Feature Correlation Module at the smallest batch sizes, where *L-SCL* consistently surpasses *SCL* in all **batch size 8 configurations** for both datasets. Additionally, comparing *LOCAL* with *L-SCL* highlights the impact of the proposed OCL loss illustrated in Eq. (2), which enforces orthonormality among negatives, while all other factors remain unchanged between the two methods.

Furthermore, we observe greater stability of our proposed method compared to *SCL*. Using the HRA dataset as an example, the stability is evident in two aspects: (a) Our method shows minimal variation

(within 1%) between batch sizes of 16 and 8 (e.g., 83.64% to 82.78% for $512 \times 512$ input), while **SCL** experiences a larger drop of around 2% (78.49% to 76.47%). (b) For 5-fold cross-validation, our method has lower variance (around 3%, ranging from 81.90% to 84.57%), compared to **SCL**'s higher variance, which reaches up to 5% (74.01% to 79.00%) in the batch size 8, $512 \times 512$ configuration.

**Performance Comparison for Distinct Classes with *Smallest* Batch Sizes and Smallest Resized Images.** Table 3 shows the categorical performance of each class under the most constrained configuration, with the smallest batch size and image resolution to simulate limited GPU memory conditions. For the HRA dataset, this configuration is 8 image pairs per batch with a resolution of $256 \times 256$, and for xBD, it is 8 pairs with a resolution of $128 \times 128$. **LOCAL** consistently outperforms **SCL** across almost all metrics. effectively handles data imbalance, especially in minority categories like "severe-damage" in HRA. Additionally, LOCAL excels at identifying tree-caused damage, a particularly challenging task where trees entangled with roofs can be mistaken for "no-damage."

| Dataset | Class | Prevalence | SCL | L-SCL | LOCAL |
|---|---|---|---|---|---|
| HRA | no-damage | 52.09% | 81.40 | 83.09 | **87.36** |
| | light-damage | 19.14% | 65.20 | 66.77 | **68.61** |
| | tarp-damage | 14.6% | 81.99 | **83.78** | 82.54 |
| | tree-damage | 9.16% | 66.44 | 69.85 | **74.43** |
| | severe-damage | 4.99% | 66.88 | 87.48 | **87.55** |
| | f1-macro | | 72.38 | 78.19 | **80.10** |
| | accuracy | | 75.87 | 78.40 | **81.76** |
| xBD | no-damage | 69.79% | 89.54 | 89.79 | **90.48** |
| | minor-damage | 11.0% | 50.40 | 49.68 | **51.68** |
| | major-damage | 12.97% | 68.65 | 67.34 | **69.48** |
| | destroyed | 6.24% | 79.01 | **80.05** | 78.90 |
| | f1-macro | | 71.90 | 71.72 | **72.53** |
| | accuracy | | 80.83 | 81.01 | **82.00** |

Table 3: Categorical performance of each class with smallest batch size and image down-scaling setting

**Visualization of Learned Embeddings.**

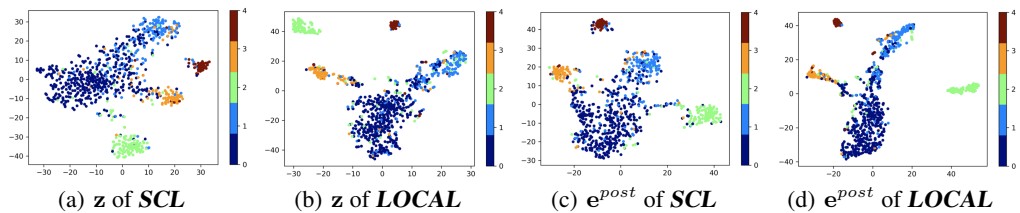

(a) z of **SCL**  (b) z of **LOCAL**  (c) $\mathbf{e}^{post}$ of **SCL**  (d) $\mathbf{e}^{post}$ of **LOCAL**

Figure 7: The t-SNE visualization on HRA test split

Fig. 7 displays the t-SNE visualization Van der Maaten & Hinton (2008) of the embedding $\mathbf{z}$ from the paired images, as well as $\mathbf{e}^{post}$ from post-disaster images in HRA. Colors indicate classes, with class numbers 0-4 representing the 5 categories of damage types, respectively. **LOCAL** leads to more separated and compact clusters compared to the embedding learned by **SCL** The "tarp-damage" (green) and "severe-damage" (red) show more purified color, indicating higher precision than **SCL**. Similar observations in Fig. 7(c) and 7(d) suggest the post-event image embedding could potentially replace the input of the classification network $Clas(\cdot)$.

**Ablation Study in Backbone Network.** Table. 4 shows the results using ResNet18, ResNet34, and ResNet50 as the backbone network $Enc(\cdot)$. **LOCAL** consistently outperforms the other two on both datasets.The proposed **LOCAL** steadily increases in performance as the backbone net-

| Dataset | BackBone | SCL | L-SCL | LOCAL |
|---|---|---|---|---|
| HRA | ResNet 18 | 77.80 (75.64-79.81) | 76.87 (75.17-80.05) | **80.18** (79.58-81.21) |
| | ResNet 34 | 78.23 (75.78-79.67) | 78.00 (76.45-79.58) | **81.37** (80.97-82.37) |
| | ResNet 50 | 78.49 (76.80-80.16) | 77.98 (76.57-80.28) | **83.64** (81.90-84.69) |
| xBD | ResNet 18 | 80.81 (79.67-81.76) | 79.72 (78.68-80.33) | **81.88** (81.29-83.27) |
| | ResNet 34 | 80.41 (79.94-80.87) | 79.87 (79.39-80.13) | **82.07** (78.54-83.98) |
| | ResNet 50 | 81.23 (80.86-81.75) | 79.57 (78.52-80.98) | **82.82** (82.06-83.94) |

Table 4: Performance of under different backbones

work capability increases, while the other two do not follow a similar pattern, possibly due to embedding oscillations caused by small batches, as discussed in Section 2.

**Ablation Study for Latent Contrastive Features.** Table 5 shows the performance with latent hierarchical features from **LOCAL** and **L-SCL** . **LOCAL** consistently outperforms **L-SCL**, indicating that the proposed loss function that encourages orthonomality contributes to a better representation. While latent regularizer to all blocks enhances the **LOCAL**'s performance, but there are no clear patterns for **L-SCL** due to the oscillation caused by the small batch size, as discussed in Fig. 3.

| Dataset | Blocks | *SCL* | *LOCAL* |
|---|---|---|---|
| HRA | 1-2-3-4 | 79.07 (76.57-80.28) | **83.64** (81.90-84.69) |
| | 2-3-4 | 78.10 (76.33-79.35) | 82.27 (81.55-83.64) |
| | 3-4 | 78.86 (77.38-79.81) | 82.04 (81.09-84.22) |
| | 4 | 77.84 (76.33-79.35) | 82.92 (80.97-84.45) |
| xBD | 1-2-3-4 | 79.57 (78.52-80.16) | 82.82 (82.06-83.94) |
| | 2-3-4 | 79.42 (78.73-79.83) | **83.01** (82.54-83.58) |
| | 3-4 | 80.46(78.55-81.28) | 82.40 (81.82-82.77) |
| | 4 | 79.10 (77.85-80.88) | 82.96 (82.34-84.01) |

Table 5: Performance of latent contrastive features extracted from different layers of encoder

## 4.2 GENERALIZABILITY ON BENCHMARK DATASETS WITH SINGLE IMAGE AS INPUT

Additionally, we have performed experiments on benchmark datasets with single image as sample such as CIFAR-10-LT and CIFAR-10-LT and iNatualist-LT, to verify the effectiveness of OCL over SCL, as shown in Table 6. The obtained results suggest that encouraging orthonormality leads to improved performance, especially with small batch sizes: employing relatively small batch sizes for training: (4, 8, 12), (32, 64, 80) and (4, 8, 16) for each of the corresponding datasets.

| Dataset | BS | SCL | | OCL | |
|---|---|---|---|---|---|
| | | Accuracy | F1-macro | Accuracy | F1-macro |
| CIFAR-10-LT | 4 | 88.25 | 67.32 | **88.79** | **71.25** |
| | 8 | 92.29 | 80.48 | **92.58** | **80.70** |
| | 12 | 92.67 | 80.90 | **93.25** | **81.42** |
| CIFAR-100-LT | 32 | 74.90 | 49.21 | **75.30** | **50.85** |
| | 64 | 77.95 | 53.43 | **78.20** | **53.55** |
| | 80 | 78.35 | 54.20 | **78.65** | 53.77 |
| iNatualist-LT | 4 | 87.51 | 65.09 | **87.73** | **70.42** |
| | 8 | 93.1 | 86.77 | **93.51** | **87.07** |
| | 16 | 93.93 | 88.99 | **94.25** | **90.02** |

Table 6: Performance on benchmark datasets

We also conduct experiments on ImageNet-LT with limited epochs (thus not fully trained). We test different small batch sizes such as 4, 8, and 16 to simulate memory constraints and evaluate how OCL performs under such conditions. OCL is expected to show better performance over SCL, particularly in small batch sizes and under short

| Dataset | BS | SCL | | OCL | |
|---|---|---|---|---|---|
| | | Accuracy | F1-macro | Accuracy | F1-macro |
| ImageNet-LT | 4 | 5.74 | 4.35 | **7.12** | **5.66** |
| | 8 | 15.58 | 13.55 | **17.55** | **15.29** |
| | 16 | 26.71 | 23.65 | **29.32** | **26.06** |

Table 7: Performance on ImageNet-LT

training durations, as OCL optimizes representation more efficiently by encouraging orthonormality, which can help even under limited epochs.

## 5 CONCLUSION

In this paper, we have addressed the challenge of SCL, to avoid the model drift (class embeddings fail to form a simplex) commonly encountered when batch size is small and class distribution is highly skewed. We introduced Latent Orthonormal Contrastive Learning (LOCAL) as a solution for classification tasks involving paired data. Instead of learning the representations of distinct classes as vertices of a regular simplex inscribed in a hypersphere, the proposed approach learns orthonormal embeddings for different classes where per-class examples are mapped to unit vectors and perpenticular to the embeddings of all examples in other classes. By a simple change to the original SCL loss function (adding an absolute value to the inner products of negatives in the denominator of Eq.(2)), we are able to completely revamp the embeddings of different classes to be in orthogonal subspaces. The resultant embeddings, as tested on high resolution remote sensing imagery and natural language inference, show more discriminative power for classification. Our theoretical analysis shows that the proposed loss function has a lower bound and can actually attain its minimum without contingency on data balance unlike the standard contrastive learning. Furthermore, by incorporating the latent hierarchical correlated features via a backbone network, it allows us to further operate on small batches of paired inputs, thereby reducing memory burden.

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

## A APPENDIX

### A.1 DETAILED PROOFS

In this section, we provide the detailed proofs of the manuscript.

**Theorem A.1.** *Let $Z = \{\mathbf{z}_i \mid \forall i \in [N], \mathbf{z}_i \in \mathcal{R}^{D_E}\}$ be set of embedding features of $N$ points, and the corresponding label set is given as $Y = \{y_i \mid \forall i \in [N], y \in [K]\}$. For a fixed batch size $|B|$, we define a set of sub-sampling index sets of size $|B|$ as $\mathcal{B}$ such that*

$$\mathcal{B} = \{\{n_1, n_2, \ldots, n_B\} \mid n_i \in [N], \forall i \in [B]\}.$$

*We have*

$$\mathcal{L}_{OCL}(Z; Y) \geq \sum_{l=2}^{|B|} l M_l \log\left(l - 1 + \frac{|B| - 1}{e}\right) \tag{7}$$

*where $M_l = \sum_{y \in [K]} |\{B \in \mathcal{B} \mid |B_y| = l\}|$, and the set $B_y$ consists of all samples with label $y$ in the batch $B$. Equality is attained if and only if there are $K$ orthonormal vectors $\xi_1, \xi_2, \ldots, \xi_K \in R^{D_E}$ with a large $D_E$, s.t. $K < D_E$ can be obtained under the condition that $\forall n \in [N] : \mathbf{z}_n = \xi_{y_n}$.*

Several steps are presented in order to prove Theorem A.1 as follows.

Step 1: First let us define $B_y^C$ to be the complementary set of $B_y$ such that $B_y + B_y^C = B$. For any class $y$ and any batch $B \in \mathcal{B}$, the class-specific loss $\mathcal{L}_{OCL}(Z; Y, B, y)$ can be bounded by

$$
\begin{aligned}
&\mathcal{L}_{OCL}(Z; Y, B, y) \\
&\geq |B_y| \log(|B_y| - 1 + |B_y^C| \exp(S(Z; Y, B, y)))
\end{aligned}
\tag{8}
$$

where function $S$ can be defined as

$$S(Z; Y, B, y) = S_{att}(Z; Y, B, y) + S_{rep}(Z; Y, B, y) \tag{9}$$

In Eq. equation 10, we further introduce the two functions $S_{att}()$ and $S_{rep}()$ respectively below

$$
\begin{aligned}
S_{att}(Z; Y, B, y) &= -\frac{1}{|B_y|(|B_y| - 1)} \sum_{i \in B_y} \sum_{j \in B_y \setminus \{\{i\}\}} \langle \mathbf{z}_i, \mathbf{z}_j \rangle \\
S_{rep}&(Z; Y, B, y) \\
&= \begin{cases} \frac{1}{|B_y||B_y^C|} \sum_{i \in B_y} \sum_{j \in B_y^C} |\langle \mathbf{z}_i, \mathbf{z}_j \rangle|, & if |B_y| \neq |B| \\ 0, & if |B_y| = |B| \end{cases}
\end{aligned}
\tag{10}
$$

**Lemma A.2.** *For any class $y$ and any batch $B \in \mathcal{B}$, the class-specific loss $\mathcal{L}_{OCL}(Z; Y, B, y)$ can be bounded by*

$$
\begin{aligned}
&\mathcal{L}_{OCL}(Z; Y, B, y) \\
&\geq |B_y| \log(|B_y| - 1 + |B_y^C| \exp(S(Z; Y, B, y)))
\end{aligned}
\tag{11}
$$

*where equality holds iff all of the following hold:*

*(A1) $\forall i \in B$ there is a $C_i(B, y)$ such that $\forall j \in B_y \setminus \{\{i\}\}$, $\langle \mathbf{z}_i, \mathbf{z}_j \rangle = C_i(B, y)$.*

*(A2) $\forall i \in B$ there is a $D_i(B, y)$ such that $\forall j \in B_y^C$, $|\langle \mathbf{z}_i, \mathbf{z}_j \rangle| = D_i(B, y)$.*

*Proof.*

$$
\begin{aligned}
&\mathcal{L}_{OCL}(Z; Y, B, y) \\
&= -\sum_{i \in B_y} \frac{1}{|B_{y_i}| - 1} \sum_{j \in B_{y_i} \setminus \{\{i\}\}} \log\left(\frac{\exp(\langle \mathbf{z}_i, \mathbf{z}_j \rangle)}{\sum_{k \in B \setminus \{\{i\}\}} \exp(|\langle \mathbf{z}_i, \mathbf{z}_k \rangle|)}\right) \\
&= \sum_{i \in B_y} \log\left(\frac{\sum_{k \in B \setminus \{\{i\}\}} \exp(|\langle \mathbf{z}_i, \mathbf{z}_k \rangle|)}{\Pi_{j \in B_{y_i} \setminus \{\{i\}\}} \exp(|\langle \mathbf{z}_i, \mathbf{z}_j \rangle|)^{1/(|B_{y_i}| - 1)}}\right) \\
&= \sum_{i \in B_y} \log\left(\frac{\sum_{k \in B \setminus \{\{i\}\}} \exp(|\langle \mathbf{z}_i, \mathbf{z}_k \rangle|)}{\exp((|B_{y_i}| - 1)^{-1} \sum_{j \in B_{y_i} \setminus \{\{i\}\}} |\langle \mathbf{z}_i, \mathbf{z}_j \rangle|)}\right)
\end{aligned}
\tag{12}
$$

In Eq. equation 12, we can further reorganize the numerator below.

$$\sum_{k \in B \setminus \{\{i\}\}} \exp(|\langle \mathbf{z}_i, \mathbf{z}_k \rangle|) = \sum_{k \in B_y \setminus \{\{i\}\}} \exp(\langle \mathbf{z}_i, \mathbf{z}_k \rangle) + \sum_{k \in B_y^C} \exp(|\langle \mathbf{z}_i, \mathbf{z}_k \rangle|) \tag{13}$$

Using Jensen's inequality on both sums, one can attain In Eq. equation 12, we can further reorganize the numerator below.

$$0.8 \sum_{k \in B_y \setminus \{\{i\}\}} \exp(\langle \mathbf{z}_i, \mathbf{z}_k \rangle) \overset{(A1)}{\geq} |B_y \setminus \{\{i\}\}| \exp\left( \frac{\sum_{k \in B_y \setminus \{\{i\}\}} \langle \mathbf{z}_i, \mathbf{z}_k \rangle|}{|B_y \setminus \{\{i\}\}|} \right)$$

$$0.8 \sum_{k \in B_y^C} \exp(|\langle \mathbf{z}_i, \mathbf{z}_k \rangle|) \overset{(A2)}{\geq} |B_y^C| \exp\left( \frac{\sum_{k \in B_y \setminus \{\{i\}\}} |\langle \mathbf{z}_i, \mathbf{z}_k \rangle|}{|B_y^C|} \right) \tag{14}$$

where the the equality holds if and only if

(A1) $\exists C_i(B, y)$ such that $\forall j \in B_y \setminus \{\{i\}\}, |\langle \mathbf{z}_i, \mathbf{z}_j \rangle| = C_i(B, y)$.

(A2) $\exists D_i(B, y)$ such that $\forall j \in B_y^C, |\langle \mathbf{z}_i, \mathbf{z}_j \rangle| = D_i(B, y)$.

Plugging Eq. equation 15 in Eq. equation 13, we obtain the bound of each addend as

$$0.8 \frac{\sum_{k \in B \setminus \{\{i\}\}} \exp(|\langle \mathbf{z}_i, \mathbf{z}_k \rangle|)}{\exp((|B_{y_i}| - 1)^{-1} \sum_{j \in B_{y_i} \setminus \{\{i\}\}} |\langle \mathbf{z}_i, \mathbf{z}_j \rangle|)}$$

$$0.8 \geq |B \setminus \{\{i\}\}| + |B_y^C| \exp\left( \frac{\sum_{k \in B_y^C} |\langle \mathbf{z}_i, \mathbf{z}_k \rangle|}{|B_y^C|} - \frac{\sum_{k \in B \setminus \{\{i\}\}} |\langle \mathbf{z}_i, \mathbf{z}_k \rangle|}{|B \setminus \{\{i\}\}|} \right) \tag{15}$$

So with the definition of $S(Z; Y, B, y)$, we can obtain the claimed bound

$$\mathcal{L}_{OCL}(Z; Y, B, y)$$
$$\geq |B_y| \log(|B_y| - 1 + |B_y^C| \exp(S(Z; Y, B, y))) \tag{16}$$

$\square$

**Lemma A.3.** Let $l \in \{2, \ldots, |B|\}$. For $Y \in [K]$ and $Z$, we have $L_{LOCL}(Z, Y) = \sum_{B \in \mathcal{B}} \sum_{y \in [K]} \mathcal{L}_{OCL}(Z; Y, B, y)$, we have

$$\frac{1}{M_l} \sum_{B \in \mathcal{B}} \sum_{y \in [K]} \log(l - 1 + (|B| - l) \exp(S(Z; Y, B, y)))$$

$$\geq \log\left( l - 1 + (|B| - l) \exp\left( \frac{1}{M_l} S(Z; Y, B, y) \right) \right) \tag{17}$$

where $M_l = \sum_{y \in [K]} |\mathcal{B}_{y,l}|$ and $\mathcal{B}_{y,l}$ is an auxiliary partition of $\mathcal{B}$ such that $\mathcal{B}_{y,l} = \{B_{y_i} | |B_{y_i}| = l, \forall i \in [K]\}$. The equality holds if and only if

(A3) $l = |B|$ or there exists $D(l)$ such that for every $y \in [K]$ and $B \in \mathcal{B}_{y,l}$ the values of $S(Z; Y, B, y) = D(l)$ agree.

*Proof.* Since $f(x) = \log(l - 1 + (|B| - l) \exp(|x|))$ is a convex function, using Jensen's inequality, for every $y \in [K]$ and $B \in \mathcal{B}_{y,l}$, we have

$$0.8 \frac{1}{|\mathcal{B}_{y,l}|} \sum_{B \in \mathcal{B}} \sum_{y \in [K]} f(S(Z; Y, B, y)) \overset{(A3)}{\geq} f\left( \frac{1}{|\mathcal{B}_{y,l}|} \sum_{B \in \mathcal{B}} \sum_{y \in [K]} S(Z; Y, B, y) \right) \tag{18}$$

where the equality can be obtained if and only if A3 holds. $\square$

Step 2: Next, we use the bound of $\mathcal{L}_{OCL}(Z; Y, B, y)$ derived from Lemma A.2 and Lemma A.3 to get the bound for $\mathcal{L}_{OCL}(Z, Y)$.

**Lemma A.4.** *For every $Y$ and $Z$ the orthonormal contrastive loss $\mathcal{L}_{OCL}$ is bounded by*

$$0.85\mathcal{L}_{OCL} \geq \sum_{l=2}^{|B|} lM_l \log\left(l - 1 + (|B| - l)\exp\left(\frac{1}{M_l}S(Z; Y, B, y)\right)\right) \tag{19}$$

*where the equality holds if and only if*

*(B1) $\forall n, m \in [N]$, if $y_n = y_m$, it implies $\langle z_n, z_m\rangle \equiv \eta$.*

*(B2) $\forall n, m \in [N]$, if $y_n \neq y_m$, it implies $|\langle z_n, z_m\rangle| \equiv \gamma$.*

*Proof.*

$$\begin{aligned}
\mathcal{L}_{OCL}(Z, Y) &= \sum_{B \in \mathcal{B}} \sum_{y \in [K]} \mathcal{L}_{OCL}(Z; Y, B, y) \\
&= \sum_{l=2}^{|B|} \sum_{y \in [K]} \sum_{B \in \mathcal{B}_{y,l}} \mathcal{L}_{OCL}(Z; Y, B, y) \\
&\geq \sum_{l=2}^{|B|} \sum_{y \in [K]} \sum_{B \in \mathcal{B}_{y,l}} l\log(l - 1 + (|B| - l)\exp(S(Z; Y, B, y))) \\
&\geq \sum_{l=2}^{|B|} lM_l \log\left(l - 1 + (|B| - l)\exp\left(\frac{1}{M_l}\sum_{y \in [K]}\sum_{B \in \mathcal{B}_{y,l}} S(Z; Y, B, y)\right)\right)
\end{aligned} \tag{20}$$

The first and second inequality can be attained via Lemma A.2 and Lemma A.3. The equality can be achieved if and only if (A1), (A2), and (A3) are true. It can be further proved that $(A1)\&(A2)\&(A3) \Leftrightarrow (B1)\&(B2)$.

We first prove " $\Leftarrow$".

(A1) For an arbitrary $l \in \{2, \ldots, |B|\}$, $y \in Y$, $B \in \mathcal{B}_{y,l}$ and $i \in B$, we let $j \in B_y \setminus \{\{i\}\}$, i.e., $y_j = y_i = y$. Then we have $\langle z_i, z_j\rangle = \eta = C_i(B, y)$.

(A2) For an arbitrary $l \in \{2, \ldots, |B|\}$, $y \in Y$, $B \in \mathcal{B}_{y,l}$ and $i \in B$, we let $j \in B_y^C$, i.e., $y_j = y_i = y$. Then we have $|\langle z_i, z_j\rangle| = \gamma = D_i(B, y)$.

(A3) For an arbitrary $l \in \{2, \ldots, |B| - 1\}$, $y \in Y$, and $B \in \mathcal{B}_{y,l}$, with condition (B1), $S_{att}(Z; Y, B, y) = -\eta$, and by condition (A2), $S_{rep}(Z; Y, B, y) = -\gamma$. So we have $S(Z; Y, B, y) = S_{att}(Z; Y, B, y) + S_{rep}(Z; Y, B, y) = \gamma - \eta = D(l)$.

Next, we prove " $\Rightarrow$".

(B1) We aim to prove that given $y, y'$ and $m, n, m', n' \in [N]$ with $y_m = y_n = y$ and $y_{m'} = y_{n'} = y'$, we can induce that $|\langle z_n, z_m\rangle| = |\langle z_{n'}, z_{m'}\rangle|$.

Case I:

If $y \neq y'$, we choose $l = 2$ and we specify the batch $B' = \{\{n, m, n', \ldots, n'\}\}$ with the size $b$. We can get

$$\begin{aligned}
&S(Z, Y, B', y) \\
&= S_{att}(Z; Y, B, y) + S_{rep}(Z; Y, B, y) \\
&= -\langle z_n, z_m\rangle + \frac{|\langle z_n, z_{n'}\rangle|}{2} + \frac{|\langle z_{n'}, z_m\rangle|}{2}
\end{aligned} \tag{21}$$

With (A2), we can further get $S(Z; Y, B, y) = -|\langle z_n, z_m\rangle| + |\langle z_{n'}, z_n\rangle|$. Similarly, we can specify the batch $B'' = \{\{m', n', n, \ldots, n\}\}$ with the size $b$ and we can get $S(Z, Y, B'', y = -|\langle z_{n'}, z_{m'}\rangle| + |\langle z_{n'}, z_n\rangle|)$. Combining these two equations with condition (A3), one can deduce that $|\langle z_n, z_m\rangle| = |\langle z_{n'}, z_{m'}\rangle|$.

Case II: If $y = y'$, we choose $l = 2$ and we specify the batch $B' = \{\{m, n, p, \ldots, p\}\}$ with the size $b$. Following the similar procedure in Case I, with (A2), we can further get $S(Z, Y, B', y) =$

$-|\langle z_m, z_n \rangle| + |\langle z_n, z_p \rangle|$. Similarly, we can specify the batch $B'' = \{\{m', n', p, \ldots, p\}\}$ with the size $b$ and we can get $S(Z, Y, B', y) = -\langle z_{n'}, z_{m'} \rangle + \langle z_{n'}, z_p \rangle)$. Combining these two equations with condition (A3), one can deduce that $-|\langle z_n, z_m \rangle| + |\langle z_n, z_p \rangle| = |\langle z_{n'}, z_{m'} \rangle| + |\langle z_{n'}, z_p \rangle|$.

Now, pick the batch $B_3 = \{\{z_n, z_m, p, \ldots, p\}\}$. With condition (A2), we have $|\langle z_n, p \rangle| = |\langle z_m, p \rangle|$ and thus $|\langle z_{n'}, z_{m'} \rangle| = |\langle z_n, z_m \rangle|$.

(B2) We aim to prove that given $y \neq y'$, $|\langle z_n, z_{n'} \rangle| = |\langle z_m, z_{m'} \rangle|$.

We still choose $l = 2$ and we specify two batches as $B' = \{\{n, n, n', \ldots, n'\}\}$ with the size $|B|$ and $B'' = \{\{m, m, m', \ldots, m'\}\}$ with the size $|B|$. Assuming $S_{att}(Z; Y, B, y) = -\eta$ and thus

$$
\begin{aligned}
& S(Z, Y, B', y) \\
&= -\eta + S_{rep}(Z, Y, B', y) \\
&= -\eta + \frac{1}{2(|B| - 2)} \sum_{i \in B'_y} \sum_{j \in B'_y{}^C} |\langle z_i, z_j \rangle| \\
&= -\eta + |\langle z_n, z_{n'} \rangle|
\end{aligned}
\tag{22}
$$

Similar to Eq. 22, we have $S(Z, Y, B'', y) = -\eta + |\langle z_m, z_{m'} \rangle|$. With (A3), we have $S(Z, Y, B'', y) = S(Z, Y, B', y)$ so that $|\langle z_n, z_{n'} \rangle| = |\langle z_m, z_{m'} \rangle|$.

With (A2), we can further get $S(Z; Y, B, y) = -|\langle z_n, z_m \rangle| + |\langle z_{n'}, z_n \rangle|$. Similarly, we can specify the batch $B'' = \{\{m', n', n, \ldots, n\}\}$ with the size $b$ and we can get $S(Z, Y, B'', y = -|\langle z_{n'}, z_{m'} \rangle| + |\langle z_{n'}, z_n \rangle|)$. Combining these two equations with condition (A3), one can deduce that $|\langle z_n, z_m \rangle| = |\langle z_{n'}, z_{m'} \rangle|$. $\qquad \square$

Step 3:

Now we will partition the bounding problem into two components which characterize the intra-class bound and the inter-class bound respectively. Mathematically, a decomposition can be written as

$$
\begin{aligned}
& \sum_{y \in Y} \sum_{B \in \mathcal{B}_{y,l}} S(Z; Y, B, y) \\
&= \sum_{y \in Y} \sum_{B \in \mathcal{B}_{y,l}} S_{att}(Z; Y, B, y) + \sum_{y \in Y} \sum_{B \in \mathcal{B}_{y,l}} S_{rep}(Z; Y, B, y)
\end{aligned}
\tag{23}
$$

We first address the first addend in Eq. 24 in the following lemma. And the rest of the lemmas focus on the second addend.

**Lemma A.5.** *Let $l \in \{2, \ldots, |B|\}$ and let $Z$ to be the unit vector on a unit sphere. For every $Y$ and $Z$, it holds that*

$$
\sum_{y \in Y} \sum_{B \in \mathcal{B}_{y,l}} S_{att}(Z; Y, B, y) \geq -\left( \sum_{y \in Y} |B_{y,l}| \right)
\tag{24}
$$

*where the equality is attained if and only if: (A4) $\forall m, n \in [N]$, $y_m = y_n$ implies $z_m = z_n$.*

*Proof.*

$$
\begin{aligned}
S_{att}(Z; Y, B, y) &= -\frac{1}{|B_y||B_y \setminus \{\{i\}\}|} \sum_{i \in B_y} \sum_{j \in \mathcal{B}_y \setminus \{\{i\}\}} \langle z_i, z_j \rangle \\
&\geq -\frac{1}{|B_y||B_y \setminus \{\{i\}\}|} \sum_{i \in B_y} \sum_{j \in \mathcal{B}_y \setminus \{\{i\}\}} z_i z_j \\
&= -1
\end{aligned}
\tag{25}
$$

which can be obtained by using Cauchy-Schwarz inequality. The equality holds if and only if $z_i$ and $z_j$ are identical since the $z_i$ and $z_j$ are unit vectors. So the equality condition can be written as (A4) $\forall m, n \in [N]$, $y_m = y_n$ implies $z_m = z_n$. $\qquad \square$

Now, we use Lemma 3 and Lemma 4 to prove the bound for our orthonormal supervised contrastive loss.

**Lemma A.6.** *The orthonormal contrastive loss $\mathcal{L}_{OCL}(Z,Y)$ is bounded from below by*

$$\mathcal{L}_{OCL}(Z,Y) \geq \sum_{l=2}^{|B|} lM_l \log\left(l - 1 + \frac{|B|-1}{e}\right) \tag{26}$$

*where equality is achieved if and only if there exists $\{\xi_1, \dots, \xi_Y\}$ such that the following conditions hold:*

*(C1) $\forall n \in [N]$, $z_n = \xi_{y_n}$.*

*(C2) $\{\xi_1, \dots, \xi_Y\}$ are pairwise orthonormal.*

*Proof.* Utilizing the lower bound of $S_{att}$ in Lemma A.5, we can bound the exponential term in Lemma A.4 first below

$$\sum_{y \in [K]} \sum_{B \in \mathcal{B}_{y,l}} S(Z;Y,B,y)$$

$$\geq \sum_{y \in [K]} \sum_{B \in \mathcal{B}_{y,l}} S_{att}(Z;Y,B,y) + \sum_{y \in [K]} \sum_{B \in \mathcal{B}_{y,l}} S_{rep}(Z;Y,B,y)$$

$$\geq \sum_{y \in Y} |B_{y,l}| \times (-1) + 0 \tag{27}$$

$$= -|Y| \sum_{y \in Y} |B_{y,l}|$$

where the second term $\sum_{y \in [K]} \sum_{B \in \mathcal{B}_{y,l}} S_{rep}(Z;Y,B,y) \geq 0$ and $\sum_{y \in [K]} \sum_{B \in \mathcal{B}_{y,l}} S_{rep}(Z;Y,B,y) = 0$ if and only if $\{\xi_1, \dots, \xi_Y\}$ are pairwise orthonormal and $\forall n \in [N]$, $z_n = \xi_{y_n}$. So we can further derive the bound for $\mathcal{L}_{OCL}$ as follows.

$$\mathcal{L}_{OCL}(Z,Y)$$

$$\geq \sum_{y \in [K]} \sum_{B \in \mathcal{B}_{y,l}} S(Z;Y,B,y)$$

$$\geq \sum_{l=2}^{|B|} lM_l \log\left(l - 1 + (|B| - l) \exp\left(\frac{1}{M_l} S(Z;Y,B,y)\right)\right)$$

$$\geq \sum_{l=2}^{|B|} lM_l \log\left(l - 1 + (|B| - l) \exp\left(-\frac{\sum_{y \in Y} |B_{y,l}|}{M_l}\right)\right) \tag{28}$$

$$\geq \sum_{l=2}^{|B|} lM_l \log\left(l - 1 + (|B| - l) \exp\left(-\frac{\sum_{y \in Y} |B_{y,l}|}{\sum_{y \in Y} |B_{y,l}|}\right)\right)$$

$$\geq \sum_{l=2}^{|B|} lM_l \log\left(l - 1 + \frac{|B| - l}{e}\right)$$

$$\square$$

With Lemma 5, Theorem 1 is readily attained.

## A.2    IMPLEMENTATION DETAILS OF **SCL** AND **L-SCL**

Fig. 8 shows the detailed architecture of **SCL** model, where $\mathcal{L}_{\textbf{SCL}} = \alpha\mathcal{L}_{SCL}(\mathbf{z}) + (1-\alpha)\mathcal{L}_{WCE}$.
Fig. 9 shows the detailed architecture of **SCL** model, where $\mathcal{L}_{\textbf{L-SCL}} = \alpha\mathcal{L}_{SCL}(\mathbf{z}, \mathbf{1}) + (1-\alpha)\mathcal{L}_{WCE}$.
Fig. 10 is our proposed LOCAL method.

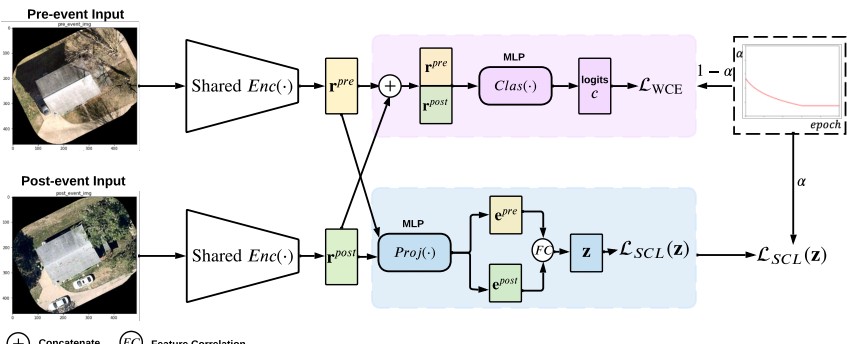

Figure 8: The end-to-end learning of the **SCL** model, $\mathcal{L}_{\textbf{SCL}} = \alpha\mathcal{L}_{SCL}(\mathbf{z}) + (1-\alpha)\mathcal{L}_{WCE}$.

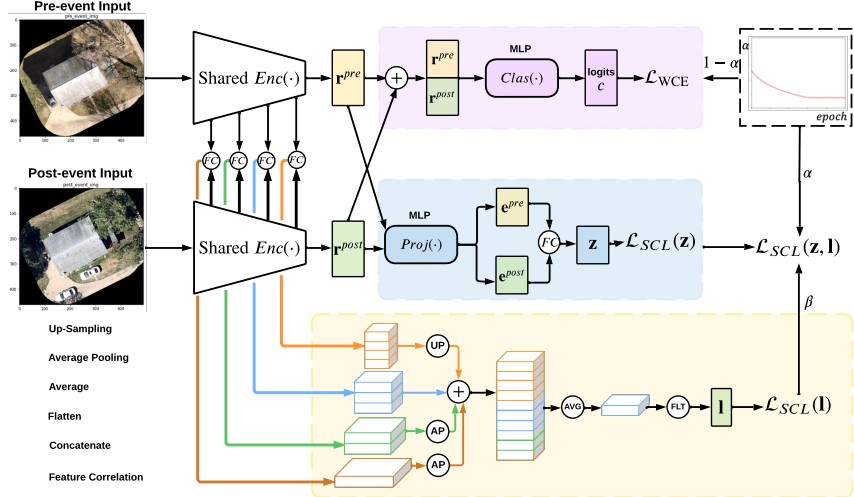

Figure 9: The end-to-end learning of the **L-SCL** model, $\mathcal{L}_{\textbf{L-SCL}} = \alpha\mathcal{L}_{SCL}(\mathbf{z}, \mathbf{l}) + (1-\alpha)\mathcal{L}_{WCE}$.

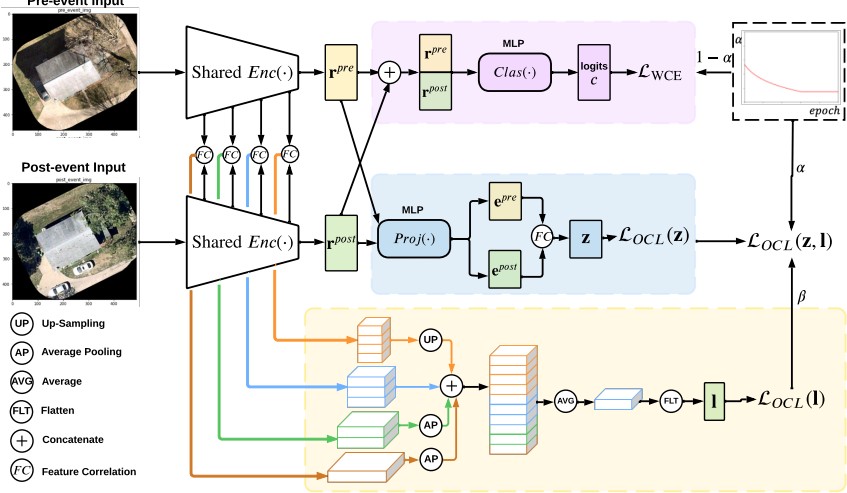

Figure 10: The end-to-end learning of the **LOCAL** model.

