# OpenReview forum: "LOCAL: Latent Orthonormal Contrastive Learning for Paired Images"
_ICLR.cc/2025/Conference — Submitted to ICLR 2025_

### Official Review · Reviewer_vcsG · 2024-11-02

**Soundness:** 2
**Presentation:** 2
**Contribution:** 2
**Rating:** 5
**Confidence:** 4

**Summary:**

This paper introduces a novel contrastive learning method aimed at addressing two issues with supervised contrastive loss: data imbalance and reliance on large batch sizes.

**Strengths:**

1. **Simplicity**: LOCAL is straightforward and easy to implement, making it accessible for practical applications.
1. **Theoretical Analysis**: The authors provide a thorough theoretical analysis of the optimization objective of LOCAL, proving a bound on the loss.
1. **Performance Improvement**: LOCAL achieves consistent performance improvements over SCL.

**Weaknesses:**

1. **Insufficient Experiments**: Although LOCAL is introduced for paired images, its applicability extends to long-tailed learning. The current experimental results significantly limit the scope of LOCAL. The paper could benefit from additional comparative experiments with other enhanced contrastive learning methods based on SCL to validate its broader effectiveness.
1. **Lack of Discussion on Related Work**: For example, there is a need to discuss methods like ProCo^[1], which also address challenges related to class imbalance and the need for large batch sizes.

[1] Probabilistic Contrastive Learning for Long-Tailed Visual Recognition. TPAMI 2024.

**Questions:**

The core idea of LOCAL involves making class representations orthogonal in latent space. However, a fixed-dimensional feature space can only accommodate a limited number of orthogonal class vectors. When the number of classes exceeds the feature dimensions, ensuring orthogonality for all class representations becomes impossible. How do the authors address this limitation, and what are the potential implications for scalability in larger class settings?

---

### Official Review · Reviewer_qKuC · 2024-11-03

**Soundness:** 3
**Presentation:** 3
**Contribution:** 2
**Rating:** 6
**Confidence:** 3

**Summary:**

This paper proposes a new contrastive learning approach to mitigate the drawback of traditional supervised contrastive learning for tasks with high-resolution data (which result to small batch size) and severe class imbalance. Specifically, it optimizes class representations in an orthonormal fashion. It conducts experiments on paired image datasets and demonstrate the superior performance of the proposed method over the traditional contrastive loss.

**Strengths:**

- The topic is interesting. The paper recognizes several drawbacks of the traditional contrastive loss when applied to tasks in satellite imagery, which has paired inputs, high memory cost and severe class imbalance, and proposes a targeted approach to these issues.
- The proposed new loss function has both theoretical and empirical validation.
- The proposed method has superior performance over baseline method on different datasets.

**Weaknesses:**

- I am a bit confused about the evaluation of the paired image dataset. What is the definition of the accuracy reported in Table 2? Do you calculate the accuracy for pre-disaster and post-disaster image together?
- The baseline compared in the paper is not thorough. The paper only considers SCL (supervised contrastive learning). It addresses the problem of class imbalance, but does not compare with methods that have been dealing with class imbalance (e.g., the papers cited in the paragraph of Line 071 in the introduction) with itself. Also, it proposes to deal with high-resolution data which will lead to high memory cost, but I'm wondering how it will compare to other memory-saving strategies for contrastive learning, e.g., a memory bank in MoCo.

**Questions:**

See weaknesses.

---

### Official Review · Reviewer_hXKz · 2024-11-04

**Soundness:** 2
**Presentation:** 2
**Contribution:** 2
**Rating:** 5
**Confidence:** 3

**Summary:**

In this paper, the authors propose a Latent Orthonormal Contrastive Learning (LOCAL) solution for paired image classification tasks. The proposed method can optimize class representation learning in an orthonormal fashion, which allows for the use of smaller mini-batches and addresses the class size imbalance. Theoretical analyses and extensive experiments demonstrate the effectiveness of the proposed method.

**Strengths:**

1. The authors introduce a novel solution of contrastive learning for paired image classification.
2. The authors conduct comprehensive experiments to demonstrate the effectiveness of the proposed method.

**Weaknesses:**

There are some grammatical errors/typos throughout the paper, which severely disturbs the readability. The reviewer recommends the authors should proofread or use a grammar checking tool to modify these typos throughout the paper. Some findings include but are not limited to:
1) Page 2, line 93, there are two “thus” in this sentence.
2) Page 3, line 152, there are two “is” in this sentence.
3) Page 5, line 243, “??” is a typo and should be modified.
4) Page 5-6, line 263 and line 272, the font color of these sentences is red. Are they typos?

**Questions:**

1. How large is a high-resolution remote sensing image? As the reviewer knows, the size of high-resolution remote sensing images has exceeded 1000 or even larger. If applicable, can the authors discuss how their method scales to larger image sizes (e.g., >1000*1000 pixels) that are common in remote sensing?
2. Can the proposed method classify pairs of non-remote sensing images? The reviewer feels the proposed method does not consider the natural characteristics of remote sensing images. If applicable, the authors should discuss potential applications or experiments with non-remote sensing images. Besides, the authors should explain what characteristics of remote sensing images the proposed method leverages, if any.
3. Why does orthonormal embedding reduce computation and use smaller mini-batches? If applicable, could the authors provide a more detailed explanation or proof of how orthonormal embeddings enable smaller batch sizes compared to standard contrastive learning approaches? One suggested way to explain it is to provide a computational complexity analysis or empirical runtime comparisons.

---

### Official Review · Reviewer_QwRj · 2024-11-05

**Soundness:** 3
**Presentation:** 3
**Contribution:** 3
**Rating:** 6
**Confidence:** 5

**Summary:**

This paper proposes a new method, Latent Orthogonal Contrastive Learning (LOCAL), for supervised contrastive learning by introducing a novel orthonormal contrastive loss, which enforces negative samples to be perpendicular to the anchor in the embedding space. This approach addresses the challenges of imbalanced classes and high computational load encountered in previous supervised contrastive learning methods when evaluated on two different pre- and post-disaster satellite image datasets.

**Strengths:**

1. Well-illustrated geometric figures in the problem statement and motivation sections for the OCL and LOCAL models.

2. The proposed OCL is supported by a theoretical analysis demonstrating that it has a lower bound and attains its minimum without contingency on data balance unlike SCL.

3. Experimental results show consistent improvement upon the evaluation tasks compared to SCL.

**Weaknesses:**

1. There are no toy experimental examples where OCL successfully optimizes but SCL definitively has an embedding drift caused by a cyclical collapse, as described in section 2 and the discussion in 3.3.

2. The HRA dataset is not cited but also is not presented as an original contribution thereby lacking sufficient context information comparable to the xBD dataset.

3. Conclusion claims to test resultant embeddings on natural language inference, but no experiments refer to natural language inference.

**Questions:**

1. Please elaborate on the procedure in the single image as sample benchmark experiment in section 4.2 and Table 6 as the models discussed are left ambiguous. Is the single image fed through the same model as described by Figure 6 (for OCL) and Figure 8 (for SCL)?

2. On all experiments in 4.1, the smallest batch size is 8. Please clarify why 8 is this minimum batch size appropriate for evaluation?

3. Discussion in 3.3 suggests a batch size large enough to enable the representation for different classes to become orthogonal is sufficient for OCL to attain a minimum. Is there a lower bound on minimum batch size (theoretically or empirically)?

4. Please compare with 'Targeted Supervised Contrastive Learning for Long-Tailed Recognition,' which provides a better baseline for addressing data imbalance in SCL.

---

### Meta-Review · Area_Chair_Lm4H · 2024-12-20

**Metareview:**

This paper introduces a new supervised contrastive learning framework, Latent Orthogonal Contrastive Learning (LOCAL), aimed at addressing two primary limitations of supervised contrastive learning (SCL): reliance on large batch sizes and challenges with imbalanced data. LOCAL introduces an orthonormal contrastive loss (OCL) that enforces orthogonality between negative samples and anchors. While the theoretical contributions and initial experimental results show promise, the paper has several critical shortcomings that undermine its contribution and applicability. The lack of robust baseline comparisons, scalability issues, and insufficient validation across diverse datasets limits its impact and relevance.

**Additional Comments On Reviewer Discussion:**

Reviewer vcsG: The scalability of LOCAL to tasks with a large number of classes is limited, and the authors provide no practical solutions or experimental evidence to mitigate this issue. The absence of comparative analysis with more advanced baselines like ProCo is a major omission.

Reviewer qKuC: The evaluation lacks clarity, particularly regarding how accuracy is calculated for paired datasets. Additionally, the failure to compare LOCAL to memory-efficient methods like MoCo diminishes the strength of the claims regarding resource efficiency.

Reviewer hXKz: The grammatical errors and inconsistencies significantly impair the paper's readability. The authors also fail to address whether LOCAL leverages unique characteristics of remote sensing images, limiting its generalizability.

Reviewer QwRj: The theoretical claims about batch size independence are not fully supported by experiments, and the experimental settings lack sufficient diversity to validate LOCAL’s robustness.

---

### Decision · Program_Chairs · 2025-01-22

Reject